# Relation between Color and Chemical Composition of Dromedary Camel Colostrum

**DOI:** 10.3390/ani13030442

**Published:** 2023-01-28

**Authors:** Halima El-Hatmi, Olfa Oussaief, Imen Hammadi, Mohamed Dbara, Mohamed Hammadi, Touhami Khorchani, Zeineb Jrad

**Affiliations:** 1LR16IRA04 Livestock and Wildlife Laboratory, Arid Land Institute of Medenine, University of Gabes, Medenine 4100, Tunisia; 2Higher Institute of Applied Biology of Medenine, Food Department, University of Gabes, Medenine 4119, Tunisia

**Keywords:** camel colostrum, color, chemical composition, correlation

## Abstract

**Simple Summary:**

Colostrum is the first milk secreted by the mammary gland of female mammals immediately after birth during the first few days. The presence of colostrum in the milk is not a good sign, especially when the milk is intended for further processing. Because the transformation of camel milk into derivative products is becoming more and more important and is gaining the interest of industries, it is therefore important to identify unsuitable milk containing colostrum for the dairy industry. The objective of this work was to study the relation between the chemical composition of camel colostrum and its color using the CIELAB color space (CIE*L** = from white to black, *a** = from red to green, and *b** = from yellow to blue). The color of colostrum reflects its quality. A paler color is associated with a lower colostrum value in terms of its general composition. Indeed, the chemical composition of camel colostrum in terms of dry matter, protein, and fat was highly correlated with the color parameters *a** (redness) and *b** (yellowness). Therefore, a color evaluation can be an effective method to detect colostrum in milk because generally there was a strong correlation between the composition of camel colostrum and its color.

**Abstract:**

Camel milk industrialization faces technological problems related to the presence of colostrum in milk. The determination of color parameters may serve to differentiate between colostrum and milk. This work aimed to study the relationship between the chemical composition of camel colostrum and milk and their colors. Samples of colostrum were collected at 2, 12, 24, 48, 72, 96, 120, 144, 168, and 360 h postpartum (*n* = 16), and their physicochemical properties (pH, acidity, viscosity, color, dry matter, ash, protein, and fat) were analyzed. The results show that all the components decreased during the first 3 days except fat. The content of this later increased from zero in the three sampling on the first day (2, 12, and 24 h) to 1.92 ± 0.61% at 48 h postpartum. The amount of total dry matter and protein decreased from 20.95 ± 3.63% and 17.43 ± 4.28% to 13.05 ± 0.81% and 3.71 ± 0.46%, respectively, during the first 7 days postpartum. There was a weak correlation between the brightness (*L**) of the camel milk and its contents of dry matter, protein, and fat; however, these parameters were strongly correlated with redness (*a**) and yellowness (*b**). Ash content was poorly correlated with the color parameters. Hence, the measurement of the color parameters of camel colostrum and milk can be a new tool to evaluate their quality.

## 1. Introduction

The colostrum, defined as the first mammary gland secretion following parturition, is characterized by its high content of immunoglobulins (Ig), mainly IgG. Dromedary camel colostrum is also a nutrient-rich fluid that is packed with immune and growth factors [1,2]. Colostrum has a nutritional profile and immunological composition that are very different from mature milk. Compared to mature milk, camel colostrum has higher protein, ash, oligosaccharide, vitamin, and mineral contents [3,4], whereas the nonprotein nitrogen and lactose contents in camel milk and colostrum showed values very close to each other [5,6]. Colostrum contains sizeable amounts of elements that function as antibacterial agents to actively promote the immune system development of a young camel [7]. 

Only a few studies have looked at how the protein composition of camel colostrum changes when it is converted to milk [1,8]. The composition of camel colostrum differs from that of conventional milk in that it contains a high concentration of whey proteins, particularly IgG, which give the newborn immunity. The deficiency of β-Lactoglobulin (β-Lg), the main whey bovine protein that causes allergy in children, and the richness in lactoferrin (LF)—an antimicrobial protein—content (an average of 2.3 g/L versus 0.5 g/L in bovine colostrum) are common characteristics of camel and human colostrums [1,2,7].

The technological aspects of milk are altered by the high levels of IgG and other soluble proteins. Numerous issues regarding bovine milk have been identified, including decreased heat stability, low cheese output, weak curd formation, and mediocre curd qualities [8]. In fact, cheese has a lower shelf life and may have off flavors when made using pasteurized or ultrahigh temperature milk that has been contaminated with colostrum [9]. For the camel milk, a similar increase in the heat sensitivity of whey proteins has been recently demonstrated [10].

Therefore, colostrum should not be present in commercial milk because of these unfavorable characteristics. However, regulations controlling the holding period of postparturition milk vary by country. This condition mostly results from the lack of a distinction between colostrum and milk; during the first postpartum week, the protein composition gradually changes from colostrum to definitive milk, and during the final weeks of pregnancy, it gradually changes again from milk to colostrum. 

Recently, there has been a great interest in camel milk processing. Many studies have been carried out to overcome the known technological difficulties during the transformation of camel milk while preserving its nutritional qualities [11,12,13,14]. 

Nowadays, various dairy products made from camel milk, including cheese, ice cream, fermented milk, coffee milk, etc., are being propagated not only in countries of the Gulf (Camelciuous), but also in Turkmenistan (Chal, Agaran, and Doiran), Mauritania (Chameaubert), Morocco (Frik), and Kazakestan (Asia) as well as in European markets. From there, industries transforming camel milk are developing in the world, and the production of heat-preserved milk is increasing as well as the production of cheese and yogurt. Cheese and pasteurized or ultrahigh temperature milk contaminated with colostrum have a shorter shelf life and may exhibit off flavors. So, because of these undesirable properties, commercial milk should be free of colostrum. This situation arises largely because there is no clear delineation between colostrum and milk; there are gradual changes in protein composition from colostrum to definitive milk during the first week postpartum. In fact, the correlation of the content of proteins, especially whey proteins, with the colostrum and its color makes it possible to detect the presence of colostrum in the milk. For this reason, the evaluation of the color parameters could be a good tool to detect colostrum in camel milk.

Indeed, the color measurement was successfully used to detect elevated milk cell contents in automatic milking systems [15] and to determine milk quality in dairy industries [16]. Gross et al. [17] reported that the color parameters clearly correlated with cow colostrum composition (IgG, fat, protein, and lactose). Up to now, no results showing the association between colostrum composition and its color have been reported for dromedary camels.

The objective of this study was to (i) specify the real time of transition from colostrum to mature camel milk and (ii) adopt a new approach to detect colostrum in camel milk. This approach consisted of studying the relationships between the chemical composition (total dry matter, protein, fat, and ash) and color attributes (*L**, *a**, and *b**) in the colostrum and milk of dromedary camels at parturition and until day 15 postpartum.

## 2. Materials and Methods

### 2.1. Sample Collection

Individual colostrum samples were obtained from the first milking postpartum (2, 12, 24, 48, and 72 h).Then, transient milk samples were collected during two days (96 and 120 h of lactation), and finally mature milk was sampled in the range of the 6–15th days (144, 168, and 360 h of lactation) of the same sixteen multiparous lactating camels (*Camelus dromedarius*) reared in Livestock and Wildlife Laboratory the Experimental Station of the Arid Land Institute of Medenine, Tunisia. Samples were obtained by manual milking, after discarding the first jets ejected, and kept frozen at –20 °C until analysis.

### 2.2. pH and Acidity

The pH was determined using a pH meter (Genway, Model 3510, Germany), and the Dornic acidity was measured by titration of 10 mL of milk or colostrum by sodium hydroxide N/9 in the presence of phenolphthalein [18].

### 2.3. Viscosity

The viscosity was expressed in centipoises (cP) and determined by applying a shear stress of 100 rpm at an oscillation frequency of 1 Hz for 1 min at 20 °C with a Brookfield-type viscometer (model DVE, MA, USA).

### 2.4. Color

Color was measured by a Chroma Meter CR-400/410 (Konica Minolta, Osaka, Japan). Camel colostrum and milk samples were transferred to a black capsule, and the color was measured immediately. Color was expressed in three scales within the visible spectrum: *L** is a light–dark that runs from 0 black to 100 white. The coordinate *a** represents red to green scale (− is green, and + is red), and *b** is blue–yellow scale (− is blue, and + is yellow).

### 2.5. Fat Analysis

This parameter was determined by the method of Gerber using butyrometer graduates [19]. This method consisted of an attack of camel colostrum or milk with sulfuric acid and separation of the fat released by centrifugation in the presence of iso-amyl alcohol.

### 2.6. Protein Analysis 

The levels of crude protein (CP) of camel colostrum and milk samples were determined by the Kjeldahl method (N × 6.38) after distillation and titration with 0.1 N hydrochloric acid [19].

### 2.7. Dry Matter and Ash Content

Dry matter, expressed in percent, was calculated after weighing the sample at 105 °C for 24 h of its dry residue. Ash content was determined after dry mineralization at 505 °C [19].

### 2.8. Statistical Analysis

The results were statistically analyzed using XLSTAT (Version 2019). To determine the significance at 5%, a one-way analysis of variance (ANOVA) followed by a Tukey’s multiple range test was employed.

The relationship between color parameters (*L**, *a**, and *b**) and chemical composition of dromedary camel colostrum and milk was determined by calculating the Pearson correlation coefficient.

## 3. Results

### 3.1. Physical Properties

#### 3.1.1. pH and Acidity

The changes in pH and acidity of camel colostrum and milk during the first 360 h of the lactation period are shown in Figure 1a,b.Three samples were taken on the first day at 2 h, 12 h, and 24 h. Then, one sample per day was taken until the seventh day and on day fifteen.

The pH value of colostrum was low (Figure 1a) in the first 24 h at 6.41 and increased to the normal value of 6.60 after 168 h of parturition (7th day). However, the acidity decreased progressively during the 360 h postpartum (Figure 2b). The highest level of acidity was detected in the first 2 h after parturition (22.02 D), whereas the lowest acidity was found on the fifteen day postpartum (15.94 D). 

#### 3.1.2. Viscosity

The evolution of viscosity of camel colostrum and milk during the first fifteen days postpartum is shown in Figure 2. The viscosity of camel colostrum decreased rapidly during the first 24 h. In fact, the viscosity diminished from 8.86 cP after 2 h of parturition to 4.75 after 12 h of parturition (nearly by half), and it reached 3.12 cP at 24 h postpartum. After that, the viscosity remained stable until day fifteen after parturition.

#### 3.1.3. Color

The variation in the color of camel colostrum and milk during 360 h after parturition was achieved by determining the average of the chromatic components (*L**, *a**, and *b**) that are represented by Figure 3a–c.

The brightness (*L**) values showed a progressive increase (97.63 ± 5.61–108.02 ± 4.38) between 2 and 168 h after parturition, and then they remained stable until hour 360 postpartum. This result shows that colostrum has a lower lightness than milk. 

Contrary to luminosity, the values of *a** gradually decreased throughout the study period, becoming negative after 168 h. The highest value of coordinate *a** (2.56 ± 0.57) was observed at 2 h postpartum, resulting in “redder” appearance of the first colostrum, while the lowest *a** value was observed at 15 days postpartum (*a** = −0.49 ± 0.23) indicating a greener color. 

Regarding the *b** values, the variation in the yellowness had the same behavior as that of the *a** values, with the only difference that the *b** values remained positive after 168 h postpartum. 

### 3.2. Chemical Composition

Changes in the gross composition (dry matter, ash, protein, and fat) of dromedary camel colostrum and milk during the 168 and at 360 h of the lactation period are shown in Figure 4a–d, respectively). 

The content of camel colostrum in the dry matter, during the first day postpartum, decreased significantly from 20.95 ± 3.63% in the first two hours to 14.22 ± 1.41% in the 24 h after parturition. Beyond the third day, the variation in the dry matter was no longer significant.

The ash content of colostrum was0.92 ± 0.11% on the first 2 h postpartum, and then it gradually decreased after 24 h to an average value of 0.84 ± 0.10 g/L. After that, it continued to decrease; however, this decrease was not significant until the fifteen day postpartum.

There was a sharp decline in the protein content from 17.43 ± 4.28 to 5.53 ± 0.91% within the first 48 h. It continued decreasing gradually to reach 4.78 ± 0.83% on the 5th day of lactation (120 h), stabilized between days 5 and 7, and further decreased to 3.89 ± 0.69 % on day 7. 

The fat content at 2 h after parturition was 0.00 %. Its content significantly increased to 1.92 ± 0.61% within the first 48 h, peaked at 3.5 ± 0.73% after 168 h, followed by a slight increase to 3.83 ± 0.56 %on day 15 postpartum. The protein and fat curves move in opposite directions.

### 3.3. Correlationbetween Color Parameters and Chemical Composition

As shown in Table 1, it is worth noting that a weak negative correlation was obtained between the *L** values and dry matter, protein, and ash contents during the 360 h postpartum. The correlation test indicated a poor positive correlation (r < 0.4) between the brightness parameter and fat composition of camel colostrum and milk. However, the correlation was negative and poor between the parameter brightness and the contents of the dry matter, protein, and ash.

Moreover, there was a strong significant positive (*p* ≤ 0.0001) correlation between the amounts of protein (%) and dry matter (%) (r = 0.837 and r = 0.7705, respectively) and the *a** values of camel colostrum and milk during the 360 h postpartum. The ash content correlated slightly with the coordinate of redness in camel colostrum and milk (r = 0.489; *p* < 0.0001). The fat content in camel colostrum and milk was negatively correlated with the *a** values. 

The correlation was positive between the *b** values and protein (r = 0.610; *p* < 0.0001) and dry matter (r = 0.567; *p*< 0.0001). The coordinate of yellowish (*b**) had a weak coefficient of correlation r < 0.4 with the ash content of camel colostrum and milk, whereas there was a high negative correlation between the amount of fat (r = –0.620) and the *b** parameter of camel colostrum and milk (Table 1).

## 4. Discussion

### 4.1. Physical Properties

Several variations occur in the physical properties of milk due to changes in its chemical composition [20].

The pH of camel colostrum and milk increased significantly between 2 h and 360 h after parturition. Likewise, Tsioulpas et al. [21] reported a significant increase in the pH of cow colostrum and milk from 6.17 in the first day postpartum to 6.58 on day 15 postpartum. Jrad et al. [5] reported lower values of pH of camel colostrum on the first day (6.28) and mature milk (6.45) than that observed in this study. The pH value of camel colostrum was lower than that of milk due to the richness of colostrum in the proteins (17.43 ± 4.28% at 2 h postpartum). El-Hatmi et al. [2] reported that the concentrations (mg/mL, means ± SD) of immunoglobulins G (IgG1, IgG2, and IgG3), camel serum albumin (CSA), and lactoferrin at the first milking were 100.7 ± 60.4, 8.5 ± 3.6, and 1.2 ± 0.3, respectively.

The acidity was high during the first day postpartum(above 19D) and dropped to reach normal levels after 6 days postpartum. The camel colostrum was slightly moreacidic than mature milk due to the buffering capacity of colostrum. These observations are in agreement with those of Jrad et al. [5], who reported a higher value of acidity in camel colostrum than in mature milk. Titratable acidity measures components that exert some buffering capacity; in addition to lactic acid, these include proteins, phosphates, citrates, and carbon dioxide [21].

The viscosity of camel colostrum was at its maximum level after 2 h postpartum. Afterward, it decreased quickly during the next 22 h. This can be attributed to the decrease in the level of dry matter. After 24 h postpartum, the viscosity was almost stable. Similar results were reported by Zhao et al. [22] for Chinese Bactrian camel colostrum and milk.

The measurement of the color parameters could be a tool to distinguish between colostrum and milk [21].The color of the samples changed with time.

The white color of milk is a result of the dispersion of reflected light by the fat globules and the colloidal particles of casein and calcium phosphate [20]. A low increase in the *L** value (lighter color) was noticed for camel colostrum over the 5 days postpartum. The *L** values of camel milk and colostrum were similar to the findings reported by Oussaief et al. [23]. However, Madsen et al. [20] and Gross et al. [17] found a higher increase in the *L** parameters of cow colostrum and milk compared to the present study. This can explained by the fact that the fat content of cow milk decreased during the first week postpartum, whereas it was completely the inverse for camel milk during this period. 

Camel colostrum, especially in the first hours after birth, is richer in lactoferrin, a protein that possess a salmonpink color, than mature milk [1,7,24], which may explain the higher redness (*a**) in colostrum. The increase in the parameter *a** is a result of the decrease in the amount of this protein during this period. 

The parameter “*b**” is related to factors associated with the milk’s natural pigment amount. For instance, milk carotenoids are responsible for the yellow color of cow milk in comparison to camel milk, which is limited in β-carotene. The probable reason for the high *b** values detected in the first hours postpartum can be that camel colostrum is rich in vitamin A [3].

### 4.2. Chemical Composition

Camel colostrum is produced during the first week and lasts 5 days according to El-Hatmi et al. [25], taking up to 2 days of transition, and then from day 7, the secretion is considered as mature milk [26].

The average content of dry matter in camel colostrum during the first 3 days postpartum decreased significantly. This drop is mainly due to the lowering of protein levels. After 3 days of parturition, the dry matter of camel colostrum and milk is almost stable; its variation was not significant. Abu-Lehia et al. [27] observed that the dry matter content of camel colostrum showed a strong decrease during the first 24 h. It continued to decrease gradually to reach a minimum value on the fifth day of lactation. Our results referring to the dry matter content in camel colostrum and milk are similar to those found by Faraz et al. [6], but are higher than those reported by Jrad et al. [5].

The ash content of colostrum is the highest on the first postpartum day; then, it gradually decreased from 0.92% to 0.73% after seven days, suggesting that camel colostrum has a higher level of minerals compared to camel milk. This result is similar to that reported by El-Hatmi et al. [25] who reported that the ash content of colostrum on the first day was 10.5 g/L and then decreased to 8 g/L after 7 days of parturition. Similarly, Jrad et al. [5] reported comparable contents of ash in camel colostrum (0.97%) and mature milk (0.75%). However, these levels are lower than those recorded by Abu-Lehia [28] who reported that the ash content of colostrum decreased slightly. Zhang et al. [8] reported a significant decrease in ash content in colostrum during the first 5 days after parturition (12.2–9.9 g/L). According to El-Hatmi et al. [25], the calcium and potassium contents were higher in colostrum than in camel milk, whereas the magnesium content did not vary between colostrum and mature milk. Konuspayeva et al. [29] reported that the mineral composition of mature camel milk was rich in potassium, sodium, chloride, and probably iron and zinc. 

The total protein in colostrum declined rapidly during the first day postpartum (from 17.43 ± 4.28 at 2 h postpartum to 8.04 ± 2.47 at 24 h *postpartum*) and during this period, consisted of greater amounts of immunoglobulins (IgGs). The high proportion of Ig in colostrum enables the young calf to develop passive immunity against common calfhood diseases. After24 h postpartum, the protein level continued to decrease until it reached 3.71 ± 0.46% at day 15 postpartum. A similar trend was observed in Najdi camel colostrum [28], where the protein content decreased from 13.00 to 5.12% within the first 24 h and further decreased to 4.02% on day 10 of lactation. Moreover, Polidori et al. [4] reported that protein contents decreased greatly from 17.2% to 4.2% during the seven days after parturition. 

The fat was almost absent during the first day postpartum. Thereafter, the fat concentration increased progressively to its average level during the first week (3.50 ± 0.73%). A similar trend was noted for dromedary camel colostrum and milk as reported by Abu-Lehia et al. [28], Faraz et al. [6], and El Hatmi et al. [25], where the fat content of colostrum was initially low, and then it reached its highest levels after about a week and then decreased to its average value thereafter. This pattern was found in human and mare colostrums but not in that of cow, buffalo, and goat. In the colostrum of these ruminants, the fat content gradually decreased starting with first day postpartum until day seven [3]. It can be theorized that the low fat content in camel colostrum is due to the ability of new calves to adapt to the physiological fluctuation of body temperature, reducing the need for calories immediately after birth. 

### 4.3. Correlation between Color Parameters and Chemical Composition

The correlation was poor between the brightness (*L**) and the chemical composition of camel colostrum and milk. Similarly, Gross et al. [17] reported a negative correlation between the *L** values and protein amounts of cow colostrum. The same observation was found for mature milk [16,30].In contrast, Gross et al. [17] found a higher correlation between *L** and the fat of cow colostrum compared to the present study. This observation is due to the richness of cow colostrum in fat. The low correlation between the parameter of color measurement *L** and the colostrum constituents (fat and protein) might be explained by their high variation during the transition from colostrum to mature milk. Unlike the parameter color *L**, there was a strong correlation between the color parameter *a** and the dry matter, protein, and fat. Camel colostrum especially in the first hours after birth is richer in lactoferrin than mature milk [1,7], which may explain the higher redness in colostrum. In contrast, Gross et al [17] stated a low correlation between the parameter of color measurement (*a**) and fat for cow colostrum. Likewise, there was a strong significant correlation between the amounts of dry matter, protein, and fat and the color parameter *b** of camel colostrum and milk during 360 h postpartum. The ash content slightly correlated with the coordinate of redness (*a**) and yellowness (*b**) in camel colostrum and milk. These results suggest that color measurements could distinguish between colostrum and milk.

## 5. Conclusions

In conclusion, these results show that big variations occur in both composition and physical properties of the mammary secretion during the transition from colostrum to milk. The physicochemical properties of colostrum differ markedly from milk, reflecting a difference in the biological function of the two secretions. The present study confirmed the relationship between camel colostrum composition and the color parameters (especially redness *a** and yellowness *b**). Camel milk industrialization faces technological problems linked to excess soluble proteins in milk if colostrum or early milk is added to the milk supplies. Color measurements of camel colostrum and milk can be a new tool to evaluate their quality in addition to laboratory analyses. A further study is necessary to study the relationship between the IgG concentration and color parameters of camel colostrum. 

## Figures and Tables

**Figure 1 animals-13-00442-f001:**
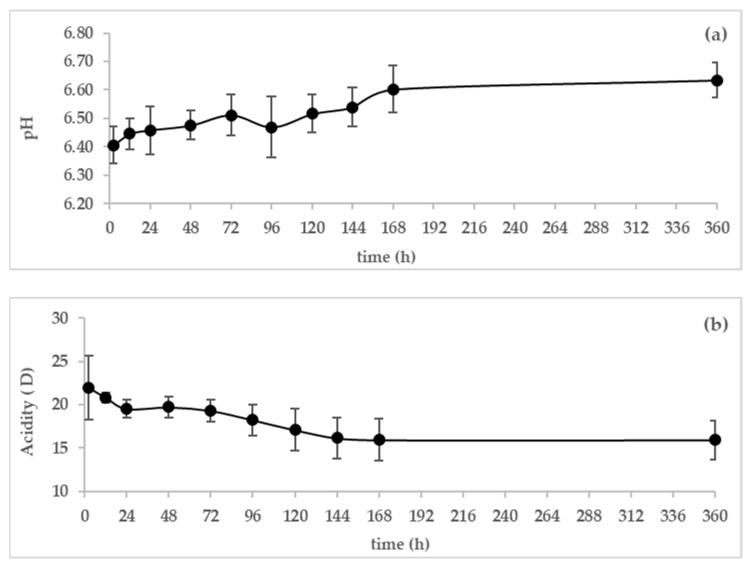
Variation in pH (**a**) and acidity (**b**) of dromedary camel colostrum and milk during the first 7 days and on day 15 postpartum (*n* = 16).

**Figure 2 animals-13-00442-f002:**
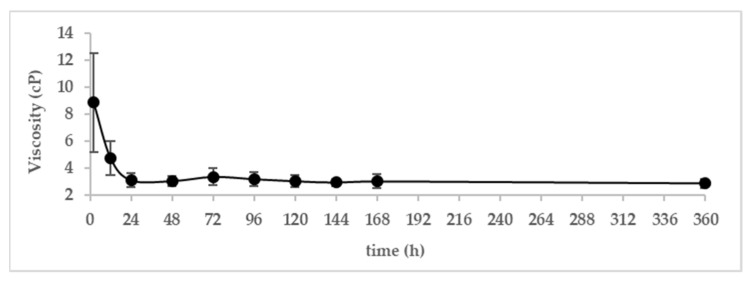
Variation in the viscosity of dromedary camel colostrum and milk during the first 7 days and on day 15 postpartum (*n* = 16).

**Figure 3 animals-13-00442-f003:**
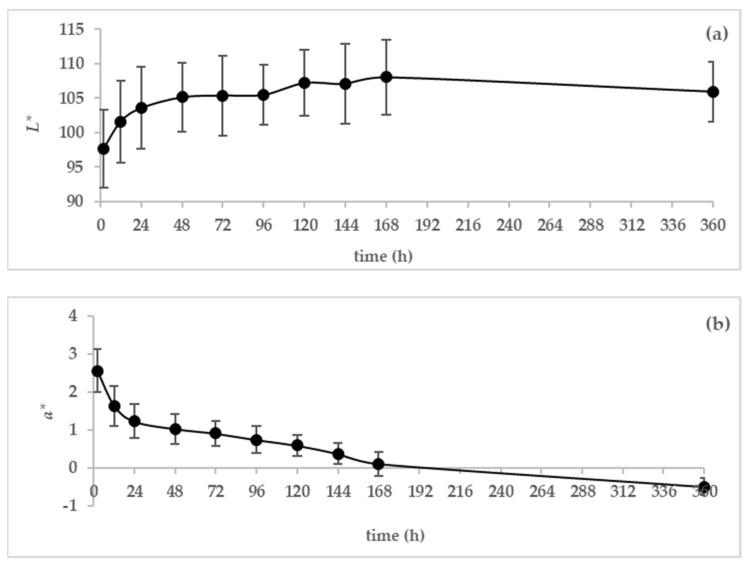
Variation in color parameters of dromedary camel colostrum and milk during the first 7 days and on day 15 postpartum (*n* = 16), (**a**) *L** (lightness), (**b**) *a** redness, and (**c**) *b** yellowness.

**Figure 4 animals-13-00442-f004:**
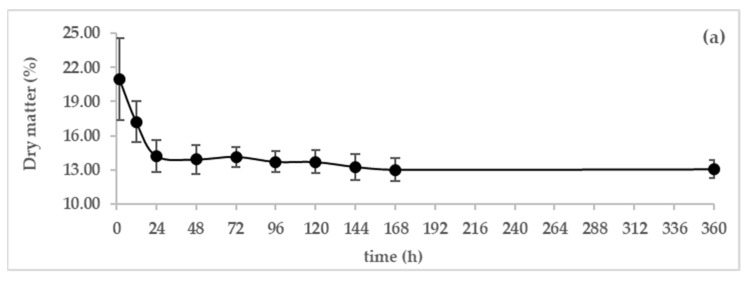
Variation in chemical composition of dromedary camel colostrum and milk during the first 7 days and on day 15 *postpartum* (*n* = 16), (**a**) dry matter, (**b**) ash and (**c**) fat, and (**d**) protein.

**Table 1 animals-13-00442-t001:** Pearson correlation coefficients (r) for dry matter, fat, protein, and ash concentrations and the CIE coordinates *L**, *a**, and *b**.

Chemical Composition	*L**		*a**		*b**	
	r	*p*-Value	r	*p*-Value	r	*p*-Value
Dry matter (%)	–0.302	0.000	0.705	<0.0001	0.567	<0.0001
Fat (%)	0.297	0.000	–0.782	<0.0001	–0.620	<0.0001
Protein (%)	–0.377	<0.0001	0.837	<0.0001	0.610	<0.0001
Ash (%)	–0.113	0.154	0.489	<0.0001	0.388	<0.0001

## Data Availability

The data presented in this study are available on request from the corresponding author.

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
