# Peer review of "Relation between Color and Chemical Composition of Dromedary Camel Colostrum"

_animals, 2023, doi:10.3390/ani13030442_

Round 1

Reviewer 1 Report

The authors of this study claim that its purpose is to investigate the connection between the chemical composition of camel milk and colostrum and their color. In the first six days of postpartum lactation, they examined samples of colostrum. Why was this time frame chosen? It is widely known that mature milk can only be collected for processing and human use at least 7-8 days’ post-partum. So what is the aim of this research?

Lines 50-51: This statement is not entirely correct. Recently, studies on the protein composition of mature milk and colostrum have been published.

For example - the colostrum composition: Nutritional Parameters in Colostrum of Different Mammalian Species by Paolo Polidori  et al. (Beverages 2022, 8, 54. https://doi.org/10.3390/beverages8030054); Camel colostrum: Nutritional composition and improvement of the antimicrobial activity after enzymatic hydrolysis by Jrad Zeineb et al. (Emir. J. Food Agric. 2015. doi:10.9755/ejfa.v27i4.19912 ).

The milk composition: Nutritional Composition and Medicinal Properties of Camel Milk, and Cheese Processing by  Muhammad Abdul Rahim et al. (International Journal of Biosciences 2020, http://dx.doi.org/10.12692/ijb/17.4.83); Characteristics and Composition of Camel (Camelus dromedarius) Milk: The White Gold of Desert by Asim Faraz et al. (Advances in Animal and Veterinary Sciences 2020, http://dx.doi.org/10.17582/journal.aavs/2020/8.7.766.7http://dx.doi.org/10.17582/journal.aavs/2020/8.7.766.77070);  Processing Challenges and Opportunities of Camel Dairy Products by Tesfemariam Berhe et al. (International Journal of Food Science 2017, https://doi.org/10.1155/2017/9061757).

In the Chapter: material and methods,  it is specified that samples were collected in the range 2-360 hours post-partum. All these are actually colostrum samples. Then it is mentioned that milk and colostrum samples were examined (when the methods were described and then when analyzing the results). What type of milk are the samples? A comparative evaluation with mature milk samples, from the same animals, would mean a collection of these milk samples after at least 7 days of lactation.

The literature cited in the article is relatively old, taking into account the dynamics of research in this field. Some examples of articles and review published in recent years, which present recent and interesting data approached by the authors of this study are mentioned above. Other recently published articles: Recent Advances in Camel Milk Processing by  Konuspayeva, G.; Faye, B. (Animals 2021, https://doi.org/10.3390/ani11041045) ; A comprehensive review on health benefits, nutritional composition and processed products of camel milk by Selva Muthukumaran et al. (Food Reviews International 2022, https://doi.org/10.1080/87559129.2021.2008953).

Author Response

Reviewer 1

** The authors of this study claim that its purpose is to investigate the connection between the chemical composition of camel milk and colostrum and their color. In the first six days of postpartum lactation, they examined samples of colostrum. Why was this time  frame chosen? It is widely known that mature milk can only be collected for processing and human use at least 7-8 days’ post-partum. So what is the aim of this research?

Authors Response:

Thank you for this relevant comment, we are agreed with the remark of the reviewer. It is well known that mature milk can only be collected for processing and human use at least 7-8 days’ post-partum. However, according to the bibliography this period varies from one species to another and from one author to another. From where, we chose to take samples during the first six days after birth in order to specify the real time of transition from colostrum to mature milk. This clarification will allow us to avoid any fraud. In fact, the correlation of the content of proteins and especially whey proteins in the colostrum makes it possible to detect the frauds of the presence of colostrum in the milk. Nowadays industries transforming camel milk are developing in the world and the production of heat-preserved milk is increasing as well as the production of cheese and yogurt. Cheese and pasteurized or ultra-high temperature milk contaminated with colostrum have shorter shelf life and may exhibit off-flavours. So, because of these undesirable properties, commercial milk should be free from colostrum. This situation arises largely because there is no clear delineation between colostrum and milk: there are gradual changes in protein composition from colostrum to definitive milk during the first week post-partum.

The aim of this study is to detect colostrum in camel milk with a very rapid method which is the colorimetry- This very rapid analysis has never been done for camel colostrum-, and this by evaluating the correlation of the gross composition and color.

** Lines 50-51: This statement is not entirely correct. Recently, studies on the protein composition of mature milk and colostrum have been published.

Authors Response:

The authors thank the referee for this comment. Recent references on the field of colostrum and milk composition were added to the revised manuscript as recommended by the reviewer (see line 51 in the revised manuscript).

** In the Chapter: material and methods,  it is specified that samples were collected in the range 2-360 hours post-partum. All these are actually colostrum samples. Then it is mentioned that milk and colostrum samples were examined (when the methods were described and then when analyzing the results). What type of milk are the samples? A comparative evaluation with mature milk samples, from the same animals, would mean a collection of these milk samples after at least 7 days of lactation.

Authors Response:

The authors thank the referee for his comments. However, the authors do not agree with the referee that samples collected in the range 2-360 hours are actually colostrum. Authors must specify that 360 h correspond to the 15th day after parturition. In fact, camel colostrum samples were collected from the first day to the third day after parturition, then the transient milk was collected during two days (the 4th and 5th day) and finally mature milk was sampling in the range 6-15th day post-partum. All samples were collected from the same animals during the fifteen days.

** The literature cited in the article is relatively old, taking into account the dynamics of research in this field. Some examples of articles and review published in recent years, which present recent and interesting data approached by the authors of this study are mentioned above. Other recently published articles: Recent Advances in Camel Milk Processing by  Konuspayeva, G.; Faye, B. (Animals 2021, https://doi.org/10.3390/ani11041045; A comprehensive review on health benefits, nutritional composition and processed products of camel milk by Selva Muthukumaran et al. (Food Reviews International 2022, https://doi.org/10.1080/87559129.2021.2008953

Authors Response:

Thank you for this relevant comment. Recent references on the field of camel milk processing were added to the revised manuscript lines 74-76.

Reviewer 2 Report

The manuscript entitled "Relation between Colour and Chemical Composition of Dromedary Camel Colostrum" is very interesting. It does possess properties that warrants publications. The objectives were well stated in the introduction, the methods are very appropriate in achieving them. The results were well presented and the discussion was adequate. The only drawback of the paper was the minor grammatical errors which were stated in the attached document.

Author Response

Reviewer 2

The manuscript entitled “Relation between Colour and Chemical Composition of Dromedary Camel Colostrum” is very interesting. It does possess properties that warrants publications. The objectives were well stated in the introduction, the methods are very appropriate in achieving them. The results were well presented and the discussion was adequate. The only drawback of the paper was the minor grammatical errors which were stated in the attached document.

Authors Response:

The authors thank the referee for these relevant comments. However, we did not find any remarks on the attached document.

Reviewer 3 Report

I accept a improved manuscript

Author Response

Thank you for giving your valuable comments.

Round 2

Reviewer 1 Report

Dear authors,

When I reviewed the revised text, I was disappointed to see that you had not changed anything or taken into account the issues and recommendations provided. You offered some interesting insights in the answer document, however none of it was used to improve the quality of the work.

Author Response

When I reviewed the revised text, I was disappointed to see that you had not changed anything or taken into account the issues and recommendations provided. You offered some interesting insights in the answer document, however none of it was used to improve the quality of the work.

Authors Response:

Thank you for this relevant comment.

  • Recent references on the field of camel milk and colostrum composition were added to the revised manuscript as recommended by the reviewer in the introduction part (see lines 51-52 and 75-77) as well as the discussion part (see lines 339-340; 347-349; 385-387; 392-393; 399-401; 409-411 and 415). These references are also added to the list of reference in the new form of manuscript.
  • The context of the study was added clearly in the introduction part (see lines 82-90 in the revised form of the paper)
  • The aim of this study has been rewritten (see lines 109-113).
  • The method of camel colostrum and milk sampling was detailed in materials and methods part (Lines 116-119).
  • The grammatical errors were corrected (lines 133; 135; 139; 142; 146 and 150).

We hope that the performed changes succeeded in improving the manuscript in a satisfactory way.

Round 3

Reviewer 1 Report

Dear authors,

I read the manuscript and noticed that you made the requested changes. The quality of the article resubmitted has increased.